# Recent Advances of Hyaluronan for Skin Delivery: From Structure to Fabrication Strategies and Applications

**DOI:** 10.3390/polym14224833

**Published:** 2022-11-10

**Authors:** Martin Juhaščik, Andrej Kováčik, Gloria Huerta-Ángeles

**Affiliations:** 1Contipro a.s., Dolní Dobrouč 401, 561 02 Dolnί Dobrouč, Czech Republic; 2Skin Barrier Research Group, Faculty of Pharmacy in Hradec Králové, Heyrovského 1203, 500 05 Hradec Králové, Czech Republic; 3Institute of Macromolecular Chemistry of the Czech Academy of Sciences, Heyrovského Nám. 2, 162 06 Prague, Czech Republic

**Keywords:** hyaluronan, self-assembly, amphiphiles, skin penetration, stratum corneum

## Abstract

Hyaluronan (HA) plays a fundamental role in maintaining the homeostasis on skin health. Furthermore, the effect of HA in skin inflammatory diseases is worth studying in the next future. HA and its conjugates change the solubility of active pharmaceutical ingredients, improve emulsion properties, prolong stability, reduce immunogenicity, and provide targeting. HA penetrates to deeper layers of the skin via several mechanisms, which depend on the macromolecular structure and composition of the formulation. The cellular and molecular mechanisms involved in epidermal dysfunction and skin aging are not well understood. Nevertheless, HA is known to selectively activate CD44-mediated keratinocyte signaling that regulates its proliferation, migration, and differentiation. The molecular size of HA is critical for molecular mechanisms and interactions with receptors. High molecular weight HA is used in emulsions and low molecular weight is used to form nanostructured lipid carriers, polymeric micelles, bioconjugates, and nanoparticles. In the fabrication of microneedles, HA is combined with other polymers to enhance mechanical properties for piercing the skin. Hence, this review aims to provide an overview of the current state of the art and last reported ways of processing, and applications in skin drug delivery, which will advocate for their broadened use in the future.

## 1. Introduction

The skin represents 16% of the total body mass (in average for a person) and fulfils a protective barrier function towards the environment [1]. On the other hand, transdermal drug delivery presents some advantages compared to the oral or parenteral routes, thereby being chosen as the uptake route for common skin treatments and long-term treatments of some chronic inflammatory skin pathologies involving atopic dermatitis (AD) [2,3], psoriasis [4], skin hyperpigmentation disorders [5,6], or cancer [7,8]. Active pharmaceutical ingredients (APIs) can be topically applied by solutions, creams, or gels for local use, improving the skin’s function and treating dermatological diseases and superficial skin tumors [8]. However, the penetration of APIs to the target sites (such as the tumor site) is hindered due to the barrier of the stratum corneum (SC), which results in limited bioavailability and poor pharmacokinetics [9]. Moreover, the hindrance to the physicochemical properties of the drug is both low solubility and low permeability. Several polymers can be used for encapsulating APIs, among them hyaluronic acid. Hyaluronic acid (hyaluronan, HA) is a biopolymer ubiquitously present in the human body and bioactive. HA consists of *D*-glucuronic acid (GlcA) and *N*-acetyl-*D*-glucosamine (GlcNAc) units and is linked by alternating *β*-(1→4)- and *β*-(1→3)-glycosidic bonds (see Figure 1) [10].

HA is the major extracellular matrix component, and 50% of total HA in the body is located in the skin [11]. HA is synthesized by enzymes located at the inner side of the plasma membrane and simultaneously exported directly to the extracellular matrix. The hyaluronan synthases (HAS) are glycosyltransferases which use as substrates UDP-α-N-acetyl-D-glucosamine (UDP-GlcNAc) and UDP-α-D-glucuronate (UDP-GlcUA) originating from cellular metabolism of glucose. HA imparts increased stability to the active targeting and boosts drug penetration. During the last years, various nano-formulations containing- HA-based drug delivery systems have been developed, e.g., niosomes [12,13], ethosomes [14,15], transfersomes [16,17,18], or polymeric micelles [19,20] and demonstrated to enhance penetration through SC (Figure 2). Furthermore, this review deals with recent advances in microneedles (MNs), as it is one of the new and more important fields, and considered one of the fastest growing markets [21].

The advantages of using HA for transdermal delivery of molecules are many, including the delivery of high molecular weight (Mw) proteins such as bovine serum albumin without degradation of their tertiary structure [22,23]. Moreover, HA is soluble in biological media, is ubiquitously present in the human body, and is biodegradable and biocompatible [10]. Furthermore, the CD44 expression on psoriatic cells could serve as a target for nanocarriers [14]; at the same time, it stabilizes the drug delivery system [15]. Epidermal HA is involved in forming an efficient epidermal barrier made of cornified keratinocytes. The organization of HA in the extracellular matrix and around cells is a critical component of the skin. HA is structured as compact pericellular coats maintained by CD44 on keratinocytes. Both HA and CD44 play an important role in normal epidermal physiological and keratinocyte functions (e.g., differentiation). The CD44 deficiency is always accompanied by a reduction in HA, and a marked alteration in keratinocyte barrier function, proliferation, differentiation, and lipid synthesis, resulting in altered skin barrier function [24]. In the case of atopic dermatitis (AD), the epidermal barrier is characterized by several alterations; HA is produced more significantly by keratinocytes than in normal skin. Epidermal HA inside AD lesioned skin is in enlarged intercellular spaces, likely due to disease-related modifications of HA metabolism. The overexpression of HAS3 causes a massive accumulation of HA in the intercellular spaces between keratinocytes, while HAS2 remains scarcely expressed in the epidermis [9].

The biological effect of HA is molecular weight-dependent. Furthermore, HA signaling properties are unique among other biologically active molecules, and depend on fragment size [25]. Indeed, high molecular weight HA (HMW-HA) with a Mw (≥10^3^ kDa) regulates the distribution of cytoskeletal proteins, preserves tissues, has immunosuppressive functions [26], and increases skin hydration [27]. Even though HMW-HA is present in homeostatic skin, fragmentation occurs under prolonged exposure to UV light or sun exposition [28]. Medium molecular weight HA (MMW-HA) is characterized by a molecular weight between 250 to1000 kDa, increases expression levels of TNF-α (tumour necrosis factor TNF), interleukin 6 and 1β, (IL-6 and IL-1 β), and chemokine ligand 2 (CCL2) in lipopolysaccharide-stimulated macrophages [29]. Low molecular weight HA (LMW-HA) is characterized by Mw from 10 to 250 kDa, activates inflammatory and immune responses, e.g., interleukins (IL-12, IL-1, IL-2, IL-4), and showed significantly higher skin penetration than HMW-HA [30]. Oligosaccharides of HA or fragments are characterized by Mw lower than 10 kDa. They promoted the differentiation of the epidermis and exhibited a more potent moisturizing effect than HMW-HA. Moreover, the HA disaccharide (ΔHA2) inhibited lipopolysaccharide (LPS)-induced inflammation. Therefore, the anti-inflammatory activity depended on HAs polymerization degree, acetyl group, and configuration [23].

## 2. The Use of Native Versus Modified Hyaluronan in Products

HA injections are one of the most common aesthetic procedures performed worldwide and represent an essential part of modern aesthetic practices, as well as in osteoarthritis treatment. However, native HA has a half-life of only 24–48 h in tissue and skin due to a fast-enzymatic degradation [31], making it unsuitable for internal use. The use of chemically modified HA improves stability, shelf-life, and viscoelasticity. Chemical derivatization and crosslinking impart appropriate mechanical properties and create an adequate macromolecular architecture. Chemical modification of HA involves the two functional sites: hydroxyl and carboxyl moieties. Furthermore, the modification can be performed at the terminal end (Figure 1). Appendix A summarizes the different reactions applied for skin delivery involving HA and includes a classification by type of reaction. Successful clinical translations of products that use modified HA are unfortunately slowed down by poor chemical characterization of bioconjugates, batch-to-batch variability, complicated multi-step manufacturing process, and inflammatory responses caused by the lack of purity of the products [32]. Nonetheless, HA is a valuable cosmetic ingredient, improving skin plumpness, increasing skin hydration, and inhibiting photoaging.

Several authors demonstrated that the immunomodulatory activities of HA could be refined, enhanced, and uncoupled from its molecular size by using chemical modification. For example, Jouy et al. described that sulphated-HA (sHA, degree of sulphation, 3.4%) exhibited a strong anti-inflammatory effect on human macrophages by decreasing IL-12, MCP-1, and IL-6. The derivative promoted the activation of regulatory functions independent of its Mw. Sulphated-HA was reported to be internalized by inflammatory macrophages mediated by CD44 [33]. Furthermore, Gao et al. reported that in contrast to conventional LMW-HA, that can trigger inflammation and immunostimulation. A butyric HA derivative demonstrated anti-inflammatory activities by modulating the cytokine expression, indicating its capacity for preventing wounds in a chronic inflammatory state [34], followed by the results of Hauck et al., who studied the role of highly-sulphated HA (sHA), HMW-HA, which served as a positive anti-inflammatory control, and LMW-HA of the same molecular size as sHA, serving as size control. All cells were stimulated with the Toll-like receptor 4 (TLR4) agonist lipopolysaccharide (LPS) to mimic a pro-inflammatory activation as it occurs in wounds. The authors detected no anti-inflammatory macrophage regulation by HMW-HA. This difference may arise from the different experimental set-ups (24 h culture with HA reported in this study versus > 48 h culture with HA in other studies) and the different cells that were used (primary murine macrophages in this study vs. primary human macrophages or murine macrophage cell lines). The results showed that sHA improved dermal wound healing in diabetic mice by reducing the inflammatory activity of cytokines and chemokines tumor necrosis factor (TNF), IL-6 and chemokine (C-C motif) ligand 2 identified as CCL2 [35].

Yuan and collaborators prepared an amphiphilic HA derivative (HA-ODA) based on LMW-HA (10 kDa) and octadecylamine by amidation. The derivative was used for coating transfersomes. These HA-modified transfersomes were prepared by thin film hydration method and membrane extrusion. The HA coating resulted in a significantly lower release rate of indomethacin. Furthermore, an enhanced transdermal effect was observed for HA-based compositions. HA probably changed the microstructure of skin layers and decreased the skin barrier function, and then the vesicle was able to squeeze through the intercellular space. The in-vivo study showed an enhanced antinociceptive effect without skin irritation [16].

## 3. The Mechanism of Skin Permeation and Retention of Hyaluronan

Even though HA is not expected to cross the SC according to the 500 Dalton rule [36], several authors described that HA crosses the SC via several mechanisms. One of the first works, Brown et al., described that HA helps the passage of low molecular weight compounds through the epidermis [37]. Today, nanoparticulated HA reaches the dermis by passive transport as demonstrated by in vitro and in vivo studies [38]. Shigefuji et al. pointed out that native HA penetrates the skin via an transcellular transport instead of the intercellular route [39].

Smejkalova et al. demonstrated that amphiphilic HA interacts with SC and enhances its permeability. The in vitro study of curcumin/Nile red-loaded micelles indicated a 3-fold higher drug deposition in the epidermis and a 6-fold higher drug deposition in the dermis after 5 h. The tracking of fluorescently labeled HA suggested a transcellular pathway for transporting amphiphilic HA loaded with Nile red [19]. Furthermore, the mechanism of penetration was studied in vitro. In this case, Nile Blue was covalently linked to HA and tracked the carrier loaded with curcumin (CUR). The effect of HA derivatized either by hexanoic or oleic acid (HA-C6 and HAC18:1, respectively) was evaluated on membrane fluidity. The membrane was fluorescently labeled by cholera toxin-Alexa647, and changes in membrane fluorescence recovery after photobleaching (FRAP method) were observed. Cells did not exhibit a significant difference in membrane fluidity of HA or free fatty acid (dissolved in DMSO) as compared to the control. Nonetheless, a significant change in fluidity was found for these derivatives. The uptake of CUR from HAC6 and HAC18:1 carrier by fibroblasts and by keratinocytes is rapid (<1 min). Interestingly, the carriers entered the cells by using active and passive transport depending on the macromolecular structure and the type of cells. The HA-C18:1-based carrier showed a great affinity to the cell membrane. It significantly changed its fluidity, following a passive transport driven by clathrin-mediated endocytosis and macropinocytosis independent of CD44, while HA-C6 prefers an active transport [40]. The authors mentioned that the comparison to other HA-containing delivery systems is rather difficult as hyalurosomes and ethosomes use HA as a surface coating to ensure CD44 targeting.

### 3.1. In Vitro and Ex Vivo Studies

In vitro measurements can be used to predict skin absorption in vivo, if properly conducted (See Table 1). The European medicine agency provided the guidelines for the evaluation of topical products [41]. Following the OECD description, “guidance document for the conduct of skin absorption studies” [42,43] is always recommended for the proper performance of experiments. Remarkably, the guidance describes vertical Franz diffusion cells, the experiment temperature, and receptor fluid composition, which are crucial for the reproducibility of the obtained data and its comparability. Several techniques such as confocal laser scanning microscopy [38], Fourier transform infrared spectroscopy (FTIR) [22], and Raman spectroscopy [44] help the analysis of the penetrated substance to receptor phase of the Franz cell and evaluate the penetration efficiency and mechanism [45]. Unfortunately, these methods are not standardized measurements. Certain invasive techniques are too non-specific or traumatic. Thus, they are generally not preferred for routine use in humans. However, a few feasible methods, most used for the pharmacokinetic assessment of topically formulations, are discussed in this review. Franz diffusion cells have proven to be a valuable tool for studying the percutaneous absorption and pharmacokinetics of applied drugs. Furthermore, vertical Franz diffusion cells were approved by the Food and Drug Administration (FDA). The model uses ex vivo, human, or animal model skin mounted in specially designed Franz diffusion cells allowing temperature and humidity that match typical in vivo conditions. These models can be artificial membranes, reconstructed skin (reconstructed human epidermis, living skin equivalents), human (ex vivo, in vivo), or animal skin [46].

Many animal models are alternatives to human skin for evaluating percutaneous penetration. The model includes pig, mouse, rat, guinea pig, and snake skins [41]. On the other hand, the human skin is the most relevant membrane model human, but the experiments are affected by ethical and regulatory obstacles. Significant variability of skin permeability could be observed between different skin donors and anatomical districts of the proband. Human skin is characterized by thickness from 60 to 100 μm (up to 600 μm in the plantar and palmar regions), while the pigs’ SC varies from 20 to 26 μm [55].

Another critical factor during the experimentation is the temperature. The skin permeation experiments should be maintained at the skin temperature of 32 ± 1 °C. However, several authors have used the body temperature, though the lipids bilayer on the skin changes its structure at 37 °C [56].

Xie et al. studied the transdermal drug permeation of HA-based ethosomes across rat dorsal skin using Franz diffusion cells. Alcohol-free rhodamine B (RB) solution and aqueous RB solution were used as the control groups, and PBS as the receptor isotonic solution. After 8 h, only a small amount of RB penetrated through the skin in the control groups. In contrast, the cumulative drug permeation of both HA-ES-RB and ES-RB was significantly higher. HA-ES-RB and ES-RB penetrated the skin through SC after 4 h, and the penetration effect of HA-ES-RB was significantly higher than ES-RB, indicating the advantage of HA in the formulation. In other words, HA improved the transdermal penetration of ethosomes. The efficient transdermal delivery of HA-ES was explained as a skin surface’s moisturizing effect on the SC [15].

Kozaka et al. performed an in vitro skin permeation study of a reverse micelle. Mouse-ear auricles were used to elucidate how fluorescently-labelled-HA (FC-HA) permeated through the epidermis and dermis. The disruption of the skin barrier function was confirmed by FTIR, where shifts in the CH_2_ symmetric stretching (~2850 cm^−1^) and the CH_2_ asymmetric stretching (~2920 cm^−1^) were observed. Thus, skin barrier lipids changed their conformation to gauche instead of physiological all-trans conformation. Furthermore, the authors used Yucatan micropig skin treated with FC-HA dissolved in PBS or the FC-HA-loaded reverse micelle formulation to study the mechanism of permeation. The green fluorescence was primarily observed in the rims of the corneocytes, which demonstrated that FC-HA followed an intercellular route. However, as the skin appendages occupy only 0.1% of the skin’s surface, trans-appendageal penetration was not dominant in this case, but it might also be present [47].

De Oliveira et al. studied the skin permeation profiles of liposomes surface’ modified with HA or HA-Chol using dermatomed pig skin mounted in Franz diffusion cells. The presence of the HA on the liposome surface was considered essential for the efficient penetration of the liposomes in the skin. The in vitro skin permeation experiments were conducted with intact skin. The amount of permeated drug through SC, epidermis, dermis, and in the receptor medium was quantified by HPLC after 16 h. The modified HA improved the skin penetration of a new quinoxaline derivative identified as LSPN331 in comparison to the controls. Indeed, the drug permeation across the SC was enhanced. The liposomes coated by the amphiphilic derivative (HA-Chol) based reached the target site of action and deeper layers of the skin compared to native HA [48].

Witting et al. analyzed the effect of HA (5 kDa) in the SC by FTIR. The reduced peak areas and the peak height of Amide I and Amide II suggested that HA changed the structure of keratin from *α*-helical into a *β*-folding shape after treatment with hyalurosomes. In fact, HA acts as a permeation enhancer by combining co-transport, increased tissue hydration, and modifying the SC, while HMW-HA (100 kDa) showed interaction with SC lipids and the highest improvement in HA in drug penetration [22].

Essendoubi et al. applied Raman microimaging to monitor the skin penetration of different molecular weights’ HA. The first step was the spectral characterization of HA for detection in the skin. Transverse skin sections were performed in the second part, and spectral images were recorded. The results showed that LMW-HA passes through the SC in contrast to the impermeability of HMW-HA [44].

Shigefuji et al. investigated the lipid fluidity of the SC after the application of HA-based nanoparticles. The authors described the importance of nanoparticulation (Figure 3). The skin penetration of a fluorescently labeled HA (FL-HA) and a nanocomplexed form (FL-HANP) was investigated using stripped hairless mouse skin. In the FL-HA treated group, fluorescence was observed in the keratinocytes, whereas in the complexed FL-HANP treated group, fluorescence was observed in the SC’s intercellular lipids [39]. However, their results were inconclusive due to the limitation of the infrared imaging in ex vivo samples and the skin surface due to water absorption of the infrared light.

Henry et al. labeled HA (Mw 10 and 400–1000 kDa) with a rhenium-tricarbonyl complex Re(CO_3_)_2_ and used it as a single-core multimodal probe for imaging. The IR spectra were recorded in the 800–4000 cm^−1^ range, and maps were generated by integrating areas with a high concentration of acyl chains at 2055–2005 cm^−1^ and 2868–2838 cm^−1^, which indicated the lipid-rich layer of the SC. The authors used human skin biopsies mounted on Franz cells, wherein derivatives of two Mw were analyzed. After 24 h, LMW-HA was homogeneously distributed in the SC and the epidermis, whereas HMW-HA was more inhomogeneously distributed with spots in the SC. The authors noted the importance of a high degree of substitution for effective detection [57].

Zhang and co-workers demonstrated that the combination of *Haliclona* sp. spicules (SHSs) and liposomes produced a synergic effect that improved the skin delivery of MMW-HA (fluorescein labeled-HA, MW: 250 kDa) dramatically. The penetration of HA was studied in vitro and 23.2 ± 3.7% of HA entered the skin mounted in Franz diffusion cells after 16 h, which was 19.4 ± 3.1-fold than the formulation in PBS, which was used as control. The results showed that 86.8 ± 4.1% of HA was accumulated in the deep skin layers. The formulation’s effectiveness after the topical application was also confirmed in vivo using Bagg albino mouse (BALB/c) [58].

Tolentino and collaborators performed release studies in vitro using a modified Franz-type diffusion cell (diffusional area = 1.3 cm^2^) mounted with hydrophilic cellulose membranes. The authors compared the deposition of NPs prepared with HA or chitosan complexed with poly-L-arginine. The NPs were loaded with clindamycin. The in vitro skin penetration experiments used full-thickness porcine pre-treated in two different ways. The first one was intact skin, skin with the pilosebaceous units artificially blocked, and sebaceous skin. The authors described that none of the formulations (NPs), nor the commercial formulation Adacne Clin^®^ (control), which contained propylene glycol as a permeation enhancer, stimulated the permeation of clindamycin across the skin as it was not detected in the receptor. The experiments with skin pieces with pilosebaceous structures artificially sealed evidenced the trans-appendageal pathway of NPs, as described before [59]. Furthermore, HA-based NPs showed 77% of drug in the SC, while chitosan-based NPs showed 50% clindamycin deposition in pilosebaceous structures [49].

Yuan et al. studied the transdermal drug permeation of indomethacin (IND) encapsulated in HA-modified transferosomes (IND-HA). The penetration across porcine ear skin was quantitatively studied using Franz diffusion cells. IND formulated in a gel was used as a control (IND/Gel). The cumulative drug permeation (Qt, μg/cm^2^) through the skin increased with time for all formulations. After 48 h of penetration, the Q values of IND/Gel, or indomethacin formulated in transferosomes in a gel (IND-Ts/Gel), and the formulation of HA modified transferosomes with indomethacin (IND-HA/Gel) were 56.54 ± 6.43 μg/cm^2^, 99.09 ± 31.58 μg/cm^2^, and 171.73 ± 30.29 μg/cm^2^, respectively. The cumulative drug permeation of IND-HA/Gel was significantly higher than that of IND/Gel. The enhanced percutaneous permeability was attributed to the deformable structure of transferosomes, which efficiently penetrate through the small channels of the skin. The surfactants incorporated in the transferosomes might have increased the lipid fluidity decreasing its packing density in the SC [16]. The lipid fluidity might have changed as the experiment was performed at 37 ± 0.5 °C.

Huerta and collaborators synthesized a derivative based on retinoic acid and HA (HA-atRA) by using LMW-HA (13 kDa) due to its therapeutic potential for treating acne, hyperpigmentation and photoaging. Nile red (NR) was encapsulated in the HA-atRA to study skin penetration (Figure 4). The encapsulated NR reached the dermis, while NR was only localized on the epidermis [6].

Pavlik et al. demonstrated that HA-atRA changed gene expression differently than retinoic acid using gene expression microarrays and showed lower cytotoxicity. Human-skin fibroblasts were treated, and their transcription profiles were compared with the untreated control. HA-atRA upregulated (>200% expression of the untreated control, *p* < 0.05) gene expression of 43 genes, while retinoic acid upregulated 87 genes under the same criteria. Furthermore, the derivative decreases cellular cholesterol, which could lower inflammation in chronic wounds or inflammatory diseases, which is very important in the case of acne vulgaris [60].

Pandey et al. prepared chitosan nanoparticles (CS-NPs) loaded with betamethasone (BMV). The authors performed an ex vivo skin penetration assay in Wistar albino rats. Furthermore, the nanoparticles were coated with HA towards CD44 targeting. The skin penetration assay was maintained under continuous stirring mode (600 rpm) at 37 °C in Franz diffusion cells. The incorporation of HA increased the encapsulation efficiency (EE) from 73.4 ± 6.7 to 87.4 ± 9.1%. A significant decrease in the zeta potential and an increase in the mean particle size of HA-BMV-CS-NPs (from 279 ± 12 to 554 ± 23 nm) were observed. HA decreased the permeation efficiency for BMV from 2056.5 to 1516.8 μg/cm^2^ [50].

Albash et al. studied the permeation of spironolactone (SP) through the skin of newly born rats, which were sacrificed, and then the dorsal skin was removed. The HA-enriched cerosomes (HA-ECs) were compared to SP suspension. The skin was frozen at −20 °C and mounted on a plastic dialysis tube with a permeation area of 3.14 cm^2^. HA-ECs were introduced into the donor compartment while the permeation medium was 50 mL of PBS at 37 ± 0.5 °C stirred at 100 rpm. The cumulative amounts permeated per unit area were calculated. The results showed that encapsulation of SP led to its retention, limiting its permeation compared to its suspension. The in vivo study showed no histopathological alternations in rats’ skin compared to untreated skin sections. These findings support the tolerability of HA-ECs for topical application [61].

Du et al. fabricated HA-based-MNs loaded with methotrexate (MTX) to treat psoriasis. The drug permeation was visualized by using confocal laser scanning microscopy. The penetration experiments were performed at 32 °C. A rhodamine 6G-loaded HA MN patch was applied on excised full-thickness porcine cadaver skin scope slide. The MNs attenuated epidermal hyperplasia and reduced IL-23 and IL-17. Skin inflammation efficiently decreased in the imiquimod-psoriasis skin model on BALB/c mice. Transdermal delivery of MTX lowered the dose and achieved the same therapeutic effect as oral administration with fewer side effects. Furthermore, the percutaneous MTX delivery evaded drug degradation in the gastrointestinal tract and first-pass metabolism in the liver.

Son et al. performed an ex-vivo skin permeation study to investigate the effect of self-assembled amphiphilic HA towards the enhancement of indocyanine green (ICG) skin permeation using Franz diffusion cells. The pig dorsal skin was maintained in a deep freezer before usage and was thawed to 32.5 °C along with PBS before the experiment. Buffer (non-treatment) and free ICG tests were used as controls [62].

### 3.2. In Vivo Therapeutic Effect of HA

By 2013, the European Union dedicated itself to removing animal testing for cosmetics, thereby giving greater prominence to human in vivo studies, particularly those that can be performed non-invasively. Hence, various imaging techniques such as confocal microscopy, tomography, and fluorescence studies received increased attention. In vivo studies are employed to evaluate the efficacy of the formulations on specific biological conditions that are hard to mimic in vitro, such as the anesthetic effect, induced vitiligo, and induced psoriasis.

For example, Yue et al. in vivo assessed an antinociceptive effect assessed by using the tail-flick test, which compared the effect of bupivacaine (BPV)-loaded nanostructured lipid carriers (NLCs) containing modified HA and the free drug. The anesthetic activity of BPV-loaded NLCs revealed a delayed and sustained activity compared with the other formulations, including liposomal BPV and cream (with 1 mg lidocaine and prilocaine, respectively). BPV encapsulation in NLCs guaranteed a prolonged antinociceptive effect that could be explained by the accumulation of the anesthetics into the upper skin layers, thus reducing the drug flux and creating a reservoir able to prolong residence time on the skin [51].

Zhang et al. performed an imiquimod-induced (IMQ) psoriasiform inflammation in mice for testing HA-modified ethosomes (ES) in vivo. The psoriasis area and severity index (PASI) were used to assess the inflammatory response in the auricles of mice [54]. The mRNA levels of TNF-α, IL-17A, IL-17F, IL-22, and IL-1β in the auricle skin of mice with psoriasis induced by IMQ were significantly higher than those of the normal control group. HA-coated ethosomes and ES groups showed downregulation of cytokines, while the HA-coated-ES inhibited IL-17A more strongly than ethosomes alone [14].

Yang and collaborators designed an ex vivo skin permeation test assayed on the abdominal full-thickness skin of Sprague-Dawley rats using Franz diffusion cells. In in vivo anesthesia, the antinociceptive effect was assessed by using the tail-flick test [51]. Yang et al. showed that NLCs presented a higher cumulative release of dexmedetomidine to the SC and showed stronger and longer antinociceptive effects at a lower dosage [63].

Furthermore, HA has been used for the construction of single-dose vaccines. Chiu et al. studied the encapsulation of ovalbumin (OVA) in a HA/chitosan composite. The OVA-loaded MNs were inserted into porcine cadaver skin or rat dorsal skin using a homemade applicator (approximately 10 N/patch) for 4 min. The OVA-loaded HA/chitosan MNs were applied to porcine skin first, and the antigen permeation across the cadaver skin was studied using Franz diffusion cells. Almost all OVA loaded in the HA tips was released and penetrated across the skin within 7 days. Compared with the HA tip, only approximately 35% OVA was released from the chitosan base during this period. These results showed that using hydrophilic HA as the tip material contributed to a rapid release of OVA. In contrast, slow release of antigen can be achieved by encapsulating OVA in the chitosan matrix. The last steps of the development included a pilot test in humans [64].

Zhuo et al. evaluated TEWL and dermatitis index in mice for evaluation of therapeutic efficacy of tacrolimus loaded in NPs coated by HA. The results showed that all test groups experienced progressive increase in TEWL and erythema intensity from week 1 to 8. Thus, the authors concluded that HA promoted regeneration of the SC, regenerated collagen, and provided an additional synergy with the components improving the pharmacological effect [65].

Elhalmoushy et al. studied the skin permeability and deposition of berberine (BRB) to treat vitiligo. The use of the formulation detected a significant increase of BRB permeation by 2.6-fold relative to that of a conventional gel. The authors suggested that the increased hydration of the SC could explain these results and followed a mechanism of hydrophilic pathways [66]. An in vitro assay of BRB-loaded hyalurosomes was performed and followed by an ex vivo model using full-thickness human skin. The authors conducted an in vivo study assessed in a vitiligo-induced animal model followed by biochemical, histological, and immunohistochemical assays to confirm the treatment’s efficacy and safety. A C57BL/6 mouse model was used for investigating the treatment [67].

### 3.3. Clinical Studies

Clinical studies are a phase of research to describe and gather more information about a device’s safety and effectiveness. HA dermal fillers are medical devices in the USA with stricter regulations than in Europe. The dermal fillers have been marketed as medical devices, medicinal products, or cosmetics. As HA fillers are not drugs, claims can be made with little to no evidence to substantiate them up to the new regulation established in 2021 [68]. HA fillers with various degrees of cross-linking are highly elastic and viscous compared with non-cross-linked HA. They are biologically compatible with treating lines and wrinkles and are very common in clinical trials [69,70], including severe acute local reactions [71].

Furthermore, there are other types of in vivo changes in skin biophysical parameters after using the product, such as hydration, transepidermal water loss (TEWL), sebum, pH, erythema intensity, elasticity, and color index. TEWL is the amount of water that evaporates through the skin and can be employed as a parameter to describe skin barrier function. A shift from the normal value 13 g/(h·m^2^) to high levels indicates an impaired barrier functionality. Most of the clinical studies are devoted to dermal fillers. The clinical studies deal with skin diseases such as vitiligo, facial seborrheic dermatitis, rosacea, and actinic keratosis and are conducted on a small sample of subjects (<100).

On the other hand, the information on cosmetic products is usually incomplete. In most cases, they are supported by in vitro data. Nonetheless, relevant differences among them are rarely reported. Depending on the type of HA used, three formulations may be identified: native HA, chemically modified HA (grafted or cross-linked), and a mixture of HA of different sizes. As an example, Sundaram et al. demonstrated more significant benefits of topical crosslinked HA over HMW HA and LMW-HA in reducing TEWL, retaining and redistributing water within the epidermis, maintaining skin integrity, and improving skin barrier structure and function. Furthermore, crosslinked HA was the most effective humectant and moisturizer compared to LMW-HA and HMW-HA, addressing skin aging without the side effects of retinoids and other topical agents [52].

Boen and collaborators performed a prospective, randomized, double-blind, placebo-controlled assay of a HA-containing serum and an antioxidant cream. The clinical trial was conducted by the Declaration of Helsinki and the International Conference on Harmonization. The formulation contained HA, palmitoyl tripeptide-28 and palmitoyl pentapeptide-4, green tea polyphenols, niacinamide, apple stem cells, vitamin E, caffeine, ceramides, and pea and botanical extracts. Subjects, evaluating investigators, and evaluating dermatopathologists were blinded. The randomized controlled trial showed that the serum and cream ameliorated the aspect of the skin on the neck. The formulation was well-tolerated by patients. As the cream vehicle contains ceramides and other active ingredients (glycerine, secondary antioxidants, and micronutrient sources, such as Leontopodium Alpinum Callus Culture Extract), both the active and placebo groups showed improvement in neck wrinkles and texture [72]. Thus, it is difficult to evaluate whether the improvement is due to the use of HA or a synergic effect.

Draelos and collaborators tested an HA-containing serum (50 and 10–1000 kDa) on 40 females (30–65 years of age) with Fitzpatrick skin types I–VI who exhibited photo-aging. HA serum showed statistically significant improvement in plumping, skin hydration, and skin smoothness as assessed in vivo. Clinical results were assessed using a digital facial analyzer (Mark-Vu^®^) before and after treatment and showed that the total skin hydration increased by 55% [27].

Avcil et al. proposed a formulation consisting of arginine/lysine polypeptide, acetyl octapeptide-3, palmitoyl tripeptide-5, adenosine, and seaweed extracts loaded in HA-MNs. The patches were applied to the outer corner of the right and left eye and a defined area on the volar forearm on healthy subjects with aged skin for 12 weeks. The product was well-tolerated and decreased 25% of wrinkles in the area [73].

Bhardwaj and collaborators recently provided evidence of wrinkle reduction mechanism and molecular changes using the three levels: in vivo (clinical), ex vivo (human skin explants/biopsies), and in vitro using a reconstituted human skin model. Interestingly, the authors did not find upregulation of aquaporin 3 (AQP3) gene expression, contrasting with another in vitro study. The authors found a significant wrinkle reduction after 28 and 56 days of daily application (clinical study) and showed ex vivo overexpression of collagen, elastin, and HA by day 7. The formulation was well tolerated in the case of normal, dry, oily and a combination of skin types and efficacy [53].

Tan et al. conducted a single-blinded intra-individual controlled clinical trial to treat keloids using 28 volunteers (85.7% males, 14.3% females). HA-based MNs were loaded with triamcinolone. MNs significantly reduced the number of keloids compared with the control. The reduction of scars was significantly higher when a higher dose of triamcinolone was used. Subjects were asked to compare the convenience of use, perceived efficacy, and overall preference between the MNs and previous treatment using corticosteroid injections. Half of the probands felt that conventional intralesional corticosteroid injection was more effective than MNs treatment. Despite this, most probands preferred microneedle treatment over intralesional injections due to less pain, self-application and convenience compared with intralesional injections. However, the conventional treatment requires 3–5 years for efficacy [74].

The cream Ialuset-vital^®^, containing HA and the extract of *Salvia haenkei*, was investigated in reducing symptoms of moderate atopic dermatitis. The study was a randomized, double-blinded, vehicle-controlled clinical study. Treatment efficacy was evaluated considering objective parameters (Scoring Atopic Dermatitis, SCORAD) and subjective parameters such as eczema measure and itching sensation. Through non-invasive TEWL determination, the treatment Ialuset-vital^®^ led to a significant increase in skin hydration throughout the treatment period, while the vehicle induced a more variable effect in the control group. Although the TEWL analysis showed a positive trend of Ialuset-vital^®^ compared to vehicles, no statistically significant difference has been shown [3].

Ulrich et al. explored the role of diclofenac sodium 3% in a HA based-gel as a treatment for actinic keratosis (AK), which is associated with a risk of progression to invasive squamous cell carcinoma and regarded as a marker of non-melanoma skin cancer. The formulation targeted AK pathophysiology through multiple mechanisms, including induction of apoptosis, inhibition of angiogenesis, and reduced inflammation [7].

Puviani et al. developed a cream containing HA, glycoproteins, and a peptide glycan complex. A prospective, assessor-blinded, 6-week study showed the efficacy and tolerability of the composition treating facial seborrheic dermatitis. This common skin condition mainly affects the scalp and changes the skin microbiota [75]. Another option is to use LMW-HA [76].

Rosacea is an inflammatory skin disease that leads to impaired skin barrier function and commonly involves the face [77]. In a recent study, Lee et al. showed that oligo-HAs increased CD44 expression. That interaction contributed to improving skin barrier functions, as confirmed by the increased filaggrin expression and reduction of TEWL. The authors analyzed the expression of IL-8 and TNF-α, which are increased in the skin with rosacea, using human keratinocytes cultured in the presence of the active cathelicidin-derived peptide LL-37. Oligo-HAs decreased the release of IL-8 and TNF-α [78]. Thus, LMW-HA can effectively normalize the cutaneous inflammatory response in rosacea.

Maggioni et al. reported a preliminary monocentric single-arm, non-blinded study to assess the clinical effect of the serum BK46 in relieving the main symptoms of rosacea: skin dryness, increased TEWL, redness, and abnormal vascularization. Twenty patients with mild to moderate rosacea were enrolled in the study and asked to apply the product twice daily for 56 days. Skin moisturization, TEWL, and erythema index were analyzed at time zero, and following 24 h and 14, 28, and 56 days of treatment. Serum BK46 contains potassium azeloyl diglycinate, squalane, dipotassium glycyrrhizate, *Aloe barbadensis* leaf juice, HA, polyacrylate crosspolymer-6, and xanthan gum. The application resulted in a diminished TEWL after 14 days of treatment, and the differences were statistically significant compared with time zero [79].

Bertolotti and collaborators tested the efficacy of non-cultured autologous epidermal cell grafting suspended in HA compared with HA alone for repigmenting vitiligo and piebaldism lesions at 6 months. Two identified paired lesions per patient were randomized to be treated by one of the devices. Among 38 patients screened, 36 (94.7%) patients were analyzed, corresponding to 72 lesions. For difficult-to-treat lesions, no repigmentation ≥50% was observed. For all other locations (*n* = 42), the success rate was significantly higher (*p* = 0.021) at 6 months and was maintained until 12 months [80].

Although HA and its derivates have significant applicability in cosmetics, further research in clinical trials is imperative. Even though several marketed products claim different effects, such as hydrating, regenerating, and anti-aging, these results are not fully demonstrated or may vary from subject to subject. Therefore, further consideration of aspects of HA metabolism is needed to explain various biological changes and predicted effects observed in vivo (Figure 5). A question is whether APIs such as HA must comply with good manufacturing practices in cosmetics.

## 4. Formulations of HA Used for Skin

Nanosystems have been investigated for enhanced skin permeation and included microemulsions (ME), nanoemulsions (NE), nanoparticles of various compositions including solid lipid nanoparticles (SLN), nanostructured lipid carriers (NLC), liposomes, and polymeric micelles. Previous studies showed that nanosystems can offer significant advantages such as the formulation of hydrophobic molecules, enhancing their solubility and, thus, bioavailability and prolonging its stability. Chemical stability is crucial for ensuring a product is safe for users [85]. Moreover, an adequate shelf-life is vital for a formulation. HA is found in cosmetics in concentrations ranging from 0.2 to 2%. The cosmetic compositions are enriched with vitamins, antioxidants, amino acids, peptides, or ceramides to enhance the activity and synergize with HA (Appendix A). The most common types of formulations were revised and are discussed below.

### 4.1. Nano and Microemulsions

LMW-HA or HMW-HA are typically employed in water phase of oil-in water (O/W) or water-oil-water (W/O/W) emulsions. Several works were analysed in the literature and resumed on Table 2. 

Water-insoluble substances could be solubilized in emulsions, while the role of HA is to protect actives from degradation, ensure controlled release, and enhance bioavailability. In the case of W/O emulsions, HA can create a gel core that stabilizes the particles and decreases their size. Amphiphilic HA can form a Pickering-type emulsion, where nanoparticles stabilize the interface [90]. Increasing the MW of HA increases the viscosity, giving higher spreadability and bioadhesive. Due to their low price, emulsions are probably the most used compositions in cosmetics [91]. Unfortunately, the concentration of HA in commercial products is usually undefined (Table 2 and Appendix A). The role of HA in cosmetics is the protection of actives from degradation, controlled release, and enhanced bioavailability. The encapsulation of water-insoluble drugs in the core of amphiphilic HA allows its solubilization in aqueous media. The choice of surfactant is a critical aspect of micro- and nanoemulsions development. Non-ionic surfactants, such as polysorbates and poloxamers, are generally preferred. For example, Hwang et al., prepared a (W/O/W) emulsion following a two-step emulsification process. The emulsion was formed by water, jojoba oil, sorbitan isostearate (polysorbate 20), grapefruit seed extract (1%), and *P. padus* extract (1%) and contained a high percent of HA (3%). The total polyphenol and flavonoid concentrations were 714.7 ± 0.5 mg/g and 72.1 ± 2.2 mg/g, respectively. Elastase inhibition (anti-wrinkle effect) and whitening effect were observed. Thus, the composition showed potential as a cosmetic agent [86]. However, the long term-stability was not studied.

Jacobus et al. incorporated azelaic acid (AZA), an agent to treat hyperpigmentation disorders, i.e., melasma, in an (O/W) emulsion. A mixture of sorbitan monooleate, poloxamer 407, and rice bran oil was used. In this case, dibutylhydroxytoluene (BTH), phenoxyethanol, and caprylyl glycol were used as preservatives. This formulation decreased tyrosinase activity, prolonged drug penetration, and improved skin retention compared to controls, which is the emulsion without HA and suspended AZA. The formulation containing HA overcomes the skin barrier, improving the efficacy of the drug [5]. Unfortunately, BHT has been shown to have tumor promotion effects [92]. Thus, its use is still controversial.

Kupper et al. prepared a (W/O) microemulsion using hydrolyzed collagen (2–3 or 340 kDa) and HA (<10 or ≥1000 kDa). The solubilization of the biopolymers depended on the ratio and content of surfactants: Span^®^ 80/Tween^®^ 80. The Mw of collagen influenced the droplet size, while HA increased viscosity and prolonged contact time. Using 2% HA concentration in combination with collagen created a gel core that stabilized the micelles and caused smaller particle size, which was stabilized by ethanol, propylene glycol, and surfactants [87].

Kozaka et al. prepared a reverse micellar solution based of LMW-HA (Mw: 10 kDa), isopropyl myristate, glyceryl monooleate, and isopropanol. A good advantage of the composition is that neither freeze-drying nor high-speed homogenization is needed. Furthermore, the authors attached fluorescein cadaverine to HA (10 kDa) by 1-ethyl-3-[3-(dimethylamino)-propyl]-carbodiimide (EDC)/N-hydroxysuccinimide (NHS) coupling reaction to study the penetration mechanism. The authors characterized the size and distribution of the reverse micelles. They were found to be monomodal and with an average size of 872 nm [47]. However, the chemical characterization of the derivative was not described in this work. 

### 4.2. Nanostructured Lipid Carriers

Nanostructured lipid carriers (NLCs) comprise solid and liquid lipids, surfactants, emulsifiers, and solvents. NLCs have emerged as a promising strategy for improving therapeutic dermal bioavailability. Particularly, amphiphilic HA exhibit properties of a tenside. Therefore, it can stabilize the NLC surface. Previous work indicated that NLCs showed a higher loading capability by conceiving a less organized solid lipid matrix, i.e., by blending a fluid lipid with the solid one. NLCs have been employed for systemic absorption improvement, site-specific treatment, and especially targeting delivery. Coating polymers are also considered for enhanced properties of the formulation. A previous study found that amphiphilic HA (conjugated with hydrophobic moieties) is used to form an amphiphilic stealth outer layer. An abstract of several formulations can be found on Table 3.

Yue et al. prepared amphiphilic HA in a sequence of two steps. The synthesis of the linker consists of the reaction of a pegylated diamine (NH_2_-PEG_5000_-NH_2_) with linoleic acid. The second step consisted of the amidation reaction of HA mediated by EDC and NHS towards the synthesis of HA-PEG-LOA. NLCs were prepared by using the amphiphilic derivative, which created a stable complex for the controlled release of bupivacaine. The liquid-lipid phase consists of a mixture of Compritol^®^ 888 ATO, and Precirol^®^ ATO [51]. The authors characterized the amide formation via infrared spectroscopy, but the degree substitution (DS) was not mentioned.

Yang et al. prepared amphiphilic HA by combining HA and 1,2-distearoyl-sn-glycero-3-phosphoethanolamine-*N*-[(poly-ethylene glycol)2000]-NH_2_ in a reaction mediated by dicyclohexylcarbodiimide (DCC). NLCs contained Compritol^®^ 888 ATO, glycerol monostearate (GMS), and soya lecithin and were prepared by solid diffusion method. The analgesics ropivacaine and dexmedetomidine were dissolved by heating at 80–85 °C. The aqueous phase was formed by amphiphilic HA and combined with a water-soluble derivative of vitamin E, which was used as a permeation enhancer together with polysorbate 80. The average size of the vesicles is around 100 nm and presented a negative zeta potential probably due to the presence of HA [63].

### 4.3. Liposomes

Liposomes are composed of one (unilamellar) or bilayer concentric membranes (multilamellar vesicles). Liposomes penetrate the skin by squeezing through the hydrophilic nanopores (0.7–30 nm) in the SC layers. Due to their size and hydrophobic and hydrophilic character, liposomes are promising systems for drug delivery because they favor the accumulation of the carried drug in the epidermis [98]. On the other hand, hyalurosomes are modified nanovesicles that possess the intrinsic characteristics of phospholipid nanovesicles potentiated with HA penetration properties and gelling capabilities [99].

Castangia et al., reported an in-vitro/in-vivo studies of liposomes and hyalurosomes loaded with licorice extract or raw glycyrrhizin. The licorice extract in ethanol showed a moderate antioxidant activity (55%). Moreover, empty liposomes and hyalurosomes exhibited a similar activity (~45%), which is explained by the formulation of phosphatidylcholine. Licorice extract loaded in vesicles inhibited 80% of DPPH free radicals and protected 3T3 cells against oxidative stress. Moreover, the vesicles loaded with the licorice extract inhibited 12-O-tetradecanoylphorbol-13-acetate (TPA) neutrophil infiltration and allowed epidermal regeneration, promoting the restoration of the superficial skin layer. HA improved the stability of the vesicles against aggregation and fusion, decreased the drug release and allowed targeting, favoring cell migration and the re-epithelization process. Oppositely, glycyrrhizin showed poor antioxidant properties, while its encapsulation in hyalurosomes enhanced the antioxidant properties [100].

Albash and collaborators reported spironolactone-loaded in HA-cerosomes (HA-ECs) prepared by ethanol injection. The authors studied in silico the importance of HA for binding both the drug and phospholipids in the formulation, confirmed by stability studies performed for 90 days. The encapsulation efficiency (EE%) ranged from 43.5 ± 1.5 to 96.8 ± 1.3%. The optimal formulae contained 10.5 mg ceramide III, 15 mg HA and Kolliphor^®^ RH40 as an edge activator. Transmission electron microscopy showed a mixed tubular and vesicular appearance with a size of 258.8 ± 2.9 to 459.3 ± 18.9 nm [61].

De Oliveira et al. conjugated a pentenoyl substituent to HA in a mixture of water/isopropanol. Hydrophobic thio-cholesterol substituents were introduced by thiol–ene reaction. Furthermore, cyanine5-amine (cy5-amine) and cyanine7-amine (cy7-amine) dyes were covalently bond either to HA or HA-S-Chol in a reaction mediated by 4-(4,6-Dimethoxy-1,3,5-triazin-2-yl)-4-methylmorpholinium chloride (DMTMM). Even though the reaction used high molar equivalents of DMTMM, low DS was obtained. The authors studied the different behavior of liposomes coated by native or amphiphilic HA (cholesterol grafted-HA). The liposomes were prepared using zwitterionic phospholipids: dipalmitoylphosphatidylcholine (DOPC), DOPE, and cholesterol dissolved in a mixture of ethanol/chloroform solution. The thin lipid film was hydrated in PBS to obtain large unilamellar vesicles. Furthermore, 2,3-Di-(4-methoxyphenyl)-quinoxaline was loaded. The liposomes coated by amphiphilic HA reached deeper layers of the skin and targeted the lesion site [48].

Franzé et al. conjugated dipalmitoylphosphatidylethanolamine (DPPE) by reductive amination at the terminal end of LMW-HA (4.8 kDa). The reaction occurs after solubilisation of DPPE in a mixture of methanol, dimethylsulfoxide (DMSO), and chloroform for 2 h at 60 °C. The imine was reduced by sodium triacetoxyborohydride, NaBH(OAc)_3_, at 60 °C for 96 h. The conjugate HA-DPPE was used for coating liposomes. The formulation improved nifedipine penetration favouring its transport to epidermis. The encapsulation of nifedipine increased its solubility up to 128-fold in HEPES buffer. The conjugate decreased the EE of nifedipine. The deformability of the vehicle decreased with HA content. The stiffening effect of HA was counterbalanced by adding ethanol, which stabilized the high content [101].

### 4.4. Ethosomes

Ethosomes are phospholipid-based vesicles containing short-chain alcohols or glycols. They present several advantages, including encapsulation of hydrophobic and hydrophilic drugs, skin hydration, and easy preparation. However, they have a complex pharmacokinetic profile. They present poor stability, and their preparation involves natural phospholipids of unknown purity. The role of HA in the formulation was recently studied. For example, Xie et al. prepared ethosomes containing HA (HA-ES) as a transdermal drug delivery system. The vehicles were loaded with Rhodamine B (RB). The ethosomes (size ~100 nm) were spherical and showed good dispersion. Different ES to HA mass ratios changed the particle size and zeta potential. The particle size of HA-ES-RB increased as a function of HA content from 594 nm to 916 nm. HA increased the zeta potential, which probably means that the stability of the carrier increased [15]. However, the reported value for one week is probably insufficient.

Zhang and collaborators prepared HA-based ethosomes using amphiphilic HA-DOPE for curcumin delivery. The amphiphilic derivative was formulated with hydrogenated soybean phospholipids, a PEGylated lipid (DSPE-PEG_2000_), and cholesterol dissolved in propylene glycol. The retention of curcumin formulated in HA-based ethosomes was up to 4-fold higher than curcumin dissolved in propylene glycol. Moreover, HA-DOPE increased both the encapsulation efficiency and stability of the vesicle. The molecular weight is crucial for the activity. Particularly, MMW HA (240 kDa) was used in the preparation of HA-DOPE, because HMW-HA possesses strong intramolecular and intermolecular interactions, which may result in the easy detachment of HA-DOPE from the liposomal membrane. If the MW of HA is low, the anchoring between HA and CD44 is weak, decreasing the targeting effect [14].

### 4.5. Niosomes

Niosomes are microscopic vesicles made from non-ionic surfactants (alkyl or dialkyl polyglycerol ethers or sphingolipids) and cholesterol. Wichayapreechar et al. prepared HA-coated niosomes loaded with *Centella asiatica* extract. Niosomes were prepared by the film-hydration method using cholesterol, Span^®^ 60 and Tween^®^ 60. HA-coating decreased the loading capacity and increased the particle size, but it allowed skin penetration, decreased burst release, and influenced its release [93]. However, the stability studies were insufficient, thus, it is not possible to predict the long-term stability of this formulation.

Sadeghi et al. prepared HA-based niosomes composed of cholesterol, sorbitan esters (Span^®^ 20, 40, 60), and Tween^®^ 80. HA-based niosomes showed higher antioxidant and anti-inflammatory properties and narrower particle size distribution [94].

### 4.6. Nanoparticles

The definition of nanoparticles changes from author to author. These kinds of vesicles impart several advantages to cosmetic and pharmaceutical formulations. For example, Wan et al. developed nanoparticles (NPs) made of amphiphilic HA. The amphiphilic derivative HA-cholesterol (30 kDa) was co-loaded with nicotinamide/tacrolimus for psoriasis treatment. HA-containing formulations were assayed on a psoriasis mouse model (induced by IMQ). The nanoparticles improved skin lesions superior to the commercial tacrolimus ointment FK506 [102]. However, the reduction of tacrolimus dose was not discussed in the publication.

Zhuo et al. prepared chitosan (~70 kDa, deacetylation degree = 85%)/polyvinyl alcohol (PVA) nanoparticles. The prepared nanoparticles were loaded with tacrolimus as a treatment for AD. The size increased by increasing the ratio of PVA or chitosan. The optimal formulation contained 0.2% of HA, decreased the crystallinity of the loaded compound, and increased the encapsulation efficiency up to 53.4%. The particles presented an average size of 223 ± 12 nm and a zeta potential of 49 ± 3.94 mV. The HA coating was considered as an additional barrier that restricted the drug diffusion, giving a controlled release. HA increased tacrolimus penetration by around 20% [65].

Shigefuji et al. complexed HMW-HA (1.2 MDa) and poly-L-lysine (PLL). Fluorophores were covalently attached to HA for studying skin penetration. Fluorescent-labeled HA (FL-HA) was prepared by a four-components condensation reaction of acetaldehyde, fluorescein amine, cyclohexyl isocyanide aiming at the carboxylic moiety of HA. FL-HA fluorescence was only detected on the skin surface. In contrast, the nanocomplex’s (FL-HANP) group showed deeper penetration. The labeling increased the particle size (from 92.2 ± 7.8 nm to 242.5 ± 22.7 nm) and decreased zeta potential from −29.5 ± 2.6 to −49.8 ± 3.9 [39].

Tolentino and collaborators prepared NPs by complexation of either HMW-HA (≥1000 kDa) or chitosan (≤50 kDa, deacetylation degree 75–85%) and poly-L-arginine. The increased proportion of crosslinker (sodium tripolyphosphate) to chitosan gave a higher crosslinking efficiency, which reduced the size of NPs to 417.4 ± 9.0 nm and polydispersity index (PDI: 0.46 ± 0.05), which is appropriate to pass through the follicular openings. In the case of HA, nanoparticles were obtained without a crosslinking agent and presented smaller hydrodynamic diameter and high entrapment efficiency (362.0 ± 18.7 nm and 48.3 ± 0.6%, respectively) [49]. Particularly, HWM-HA was used in the preparation of the nanoparticles. Thus, the NPs occluded the SC but increased hydration as reported before [22].

Kim and collaborators reported the synthesis of ovalbumin-conjugated HA-methacrylate (HA-MA-OVA). LMW-HA (29 kDa) was reacted with methacrylic anhydride at pH 8–9. The derivative was further oxidized with sodium periodate. The schizophyllan-methacrylate (SPG-MA) was synthesized using the same protocol. Furthermore, a hybrid hydrogel consisting of HAMA-OVA/SPG-MA was prepared by photopolymerization. The authors also demonstrated that HAMA-OVA nanogels did not penetrate the cells, while the hybrids resulted in higher penetration. The hybrid nanogel induced dendritic cells maturation marker and increased the expression of pro-inflammatory cytokine IL-6, involved in acute inflammation and pain. Rhodamine B-labeled nanogels were applied to fresh porcine dorsal skin. OVA-Rhodamine did not penetrate the epidermis and was deposited in SC and the outer skin layer. The SPGMA nanogels were also deposited on the skin surface, while HAMA-OVA and hybrid nanogels were observed in the dermis [103].

Wang et al. conjugated the skin/cell-penetrating peptide (SCP) to HA mediated by EDC/NHS. HA and poly (β-amino ester) (PAE) were covalently attached (HA-SCP-PAE). The authors loaded the amphiphilic (HA-SCP-PAE) with siRNA to enhance the delivery specificity to melanoma cells without transfection of skin cells. CD44 receptor was targeted by HA. The nanocarrier resulted in significant tumor growth inhibition and thus, the highest survival rate of model mice [104]. Unfortunately, the degree of chemical modification of HA in both products remained undefined, so it is difficult to predict a possible translation of the products.

### 4.7. Polymeric Micelles

Polymeric micelles are self-assembled nanoscopic core-shell structures formed by amphiphilic HA inside water, hold hydrophobic drugs in the core. For example, Son and collaborators grafted dodecylamine to LMW-HA (20 kDa) using EDC/sulfo-NHS. The chemical structure of the conjugate sodium dodecyl hyaluronate (HA–Do) was identified by IR and ^1^HNMR [62]. Unfortunately, the DS was undefined and low yields for the reaction were reported (~80%).

Cheng and co-workers combined nanosized micelles with composite MNs made of HA and carboxymethyl starch (unknown Mw). LMW-HA was chemically modified by grafting quercetin-dithiodipropionic acid (Mw 10, 44 kDa) to prepare amphiphilic HA. Micelle-loaded HA composite MNs were prepared by micromolding using HA and carboxymethyl starch (2:1). A higher amount of CMS-Na produced highly viscous solutions, which were impossible to be processed. Still, HA tailored the viscosity for the processing. Polymeric micelles were prepared for curcumin encapsulation. Micelles released approximately 42% of curcumin within the first hour and 74.7% after 6 h. Penetration studies were performed in vitro using self-made diffusion cells at 32 °C [20]. However, the authors used formamide for loading, a class II solvent of limited use due to its neurotoxicity and teratogenicity.

Wang Y. and collaborators recently prepared a conjugate of cyclodextrin grafted to HA to form an amphiphilic derivative identified as (HA-CD). The derivative was used for Paeonol (PAE) loading while forming the complex (HA-CD-PAE) to treat AD. The cyclodextrin cavity is hydrophobic. Thus, it can encapsulate the hydrophobic components such as PAE, which has a LogP value of PAE: 2.16, indicating its strong hydrophobicity. The particle size of HA-CD-PAE was 177.8 ± 9.19 nm (PDI 0.241 ± 0.019), and the zeta potential of −12.90 ± 0.55 mV. The morphology of HA-CD-PAE polymeric micelles was uniform and spherical shape. The EE (%) and DL (%) of PAE was determined to be 63.94 ± 0.72% and 3.84 ± 0.4 % [2]. Though the authors identified the degree of substitution as 24.8%, the signal of the amide formation was not evident in the infrared spectra.

### 4.8. Microneedles

Microneedles (MNs) are micron-scale devices/projection arrays (50–900 μm), which can painlessly penetrate the outermost SC and facilitate intradermal and transdermal delivery of APIs, vaccines or peptides [105]. They combine the benefits of both hypodermic needles and transdermal patches and are gradually changing the field of transdermal and intradermal drug delivery. The physical parameters of MNs to be considered during the development are a sharp tip and mechanically most robust, essential for piercing the skin. A second component is usually mixed with HA to ensure mechanical strength, prolonged stability, biodegradability, dissolution time, and sustained release (see Table 4 with examples of formulations containing HA) [106]. 

For example, the force required for a reliable skin penetration was ~0.058 N per needle or 5.8 N per array (100 needles) to pierce through the top skin layer without breaking [114]. They are currently progressing through clinical trials as anti-wrinkles (ClinicalTrials.gov Identifier: NCT04989361) and anti-psoriasis treatments [115]. The MNs HA-based patch Thera-Pass^®^ RMD-6.5A is already available in Korea and was relatively quickly commercialized due to bioequivalence or omission of clinical phases I and II (NCT02955576).

Jang et al. compared the clinical skin improvement effect of HMW-HA (800 kDa) to LMW-HA (39 kDa) dissolving microneedle patch. The patch contained adenosine (Ad), a known anti-ageing agent. A 12-week trial was performed on 23 healthy women of approximately 51.83 ± 2.85 years of age who had wrinkles around their eyes (the crow’s feet area). A significant improvement of skin elasticity was observed for both groups during 8 weeks of product application compared to time 0. The improvement of the skin elasticity was 7.99% and 7.35% for the Ad-HMN and Ad-LMN, respectively, after 8 weeks [107].

Fonseca and colleagues developed a pyramidal-shaped MN matrix patch for topical rutin delivery. The patch was fabricated using bacterial cellulose (BC) and HA (Mw: 400 kDa) to encapsulate rutin. The authors indicated that BC as a back layer reinforces the MNs patch. The MNs patch consisted of 225 needles with sufficient mechanical force to withstand skin insertion with a failure force higher than 0.15 N per needle [108].

Yu et al. prepared MNs based on crosslinked 3-aminophenylboronic acid-modified alginate (Alg-APBA) and HA (undefined Mw) for transdermal delivery of insulin. The height of the microneedle was determined to be ~650 μm, and the space between each two MNs is ~600 μm. A square with a side length of ~300 μm to a tip measuring ~10 μm in width. Furthermore, the failure force of Alg-APBA/HA MNs reaches 0.37 N per needle due to crosslinking [116].

Chiu et al. combined HA (7 and 200–500 kDa) with chitosan (degree of deacetylation 91.2%) and polyvinyl alcohol/polyvinylpyrrolidone. OVA was labeled with different fluorophores and encapsulated in the HA tip and chitosan base of the MN to monitor the antigen release from these two matrices. The composite MNs pierced SC and were implanted in the dermal layer, with an insertion depth of 612 ± 46 µm. The combination of HA/chitosan induced antibody responses after two weeks of immunization and significantly increased specific IgG serum antibody levels compared with chitosan alone [64].

Leone and co-workers explored the effect of Mw on the processing of HA towards the encapsulation of endotoxin-free OVA. The liquid formulations of OVA alone or mixed with different HAs were intradermally injected using hollow MNs. The authors studied the piercing and dissolution of MNs. The processing included molding, vacuum, centrifugation, drying and curing towards the fabrication of arrays. The sharpness was independent of the Mw (Figure 6), but LMW-HA (150 kDa) showed a penetration efficiency higher than 96% [109].

A recent study on structure-activity between the molecular weight of HA and transdermal delivery efficiencies of HA-based MNs showed that 74 kDa is more efficient. In that work, HAs were characterized by three molecular weights (i.e., ~10, 74 and 290 kDa) as the MN matrix for MNs, which were fabricated through a template method. The release of rhodamine B (RB) was monitored in the receptor for 48 h. HA-MN (74 kDa) exhibited the highest release amounts of RB, with a cumulative release of 96.3% over 48 h. In contrast, a smaller amount of RB was released in the 10 and 290 kDa groups [117].

Xie et al. fabricated HA-based MNs loaded with bleomycin for inhibiting hypertrophic scars. The MNs presented sufficient mechanical strength (65.4 MPa) to pierce the scarred skin ex-vivo with an insertion depth of 300 μm. The dissolution behavior of bleomycin loaded in MNs on the skin was studied in-vivo. The results showed that half of the MN shafts dissolved after 2 min, while 2/3 of the MNs dissolved within 5 min. After 10 min, almost all the shafts dissolved, and a slight dissolution of the base was observed. Probably, the fast dissolution explained the fast release of bleomycin, 20% of the loaded bleomycin was released within 1 min and 52% within 10 min. Interestingly, MN inhibited hypertrophic scar by downregulating the secretion of transforming growth factor beta 1 (TGF-*β*1) [110].

In addition, Zhu and co-workers loaded 5-aminolevulinic acid (5-ALA) in HA-based MNs. The MNs effectively penetrated SC, while the encapsulation provided an acidic and oxygen-free environment to reduce the dimerization of 5-ALA. Thus, it maintains the activity. The mechanical properties of loaded MNs slightly decreased compared to empty MNs. The 5-ALA loaded-MNs displayed tumor elimination in vitro and in vivo. The activity was kept after storage at 25 °C for nine months, making it device with great potential for photodynamic cancer therapy [8].

## 5. Conclusions and Outlook

In summary, even though low molecular weight HA penetrates the stratum corneum, the effect of molecular weight is ignored in several works. The chemical modification of HA for skin uses fragmented HA, LMW-HA, and MMW-HA with low degrees of chemical modification.

A challenge is to develop a more effective scale-up to fast the translation from the laboratory bench to the market. Most existing methods for conjugate synthesis lack reproducibility. The conjugates lack structural uniformity, leading to failure during clinical trials or problems during the regulatory process.

Emulsions are the most frequent type of hyaluronan formulation in cosmetics, but researchers usually did not consider the effect of concentration of HA, Mw, and its stability. HA increases skin hydration and elasticity and decreases transepidermal water loss. The efficacy of HA as a retardant of skin ageing is currently evaluated in clinical studies.

LMW-HA and MMW-HA are chemically modified for the preparation of amphiphilic HA. The coating uses amphiphilic HA for stabilization, and prolonged stability of vesicles is reported. The presence of HA increases spreadability, allows sustained release of drugs, improves cargo stability, and avoids drug leakage.

HA provides targeting to the CD44 receptor; Therefore, it boosts the anti-psoriatic efficacy of ethosomes. LMW-HA is preferred for the processing of microneedles. In contrast, a cosmetic formulation could incorporate high molecular weight HA or its amphiphilic derivatives for drug encapsulation.

Furthermore, it is difficult to assess the improvement of topical agents in vivo because these are subtle effects that might be difficult to capture. Moreover, there are variations between probands. Therefore, as shown in this manuscript, the efficacy of hyaluronic acid depends on the molecular weight, but it is always uncharacterized in several research works.

## Figures and Tables

**Figure 1 polymers-14-04833-f001:**
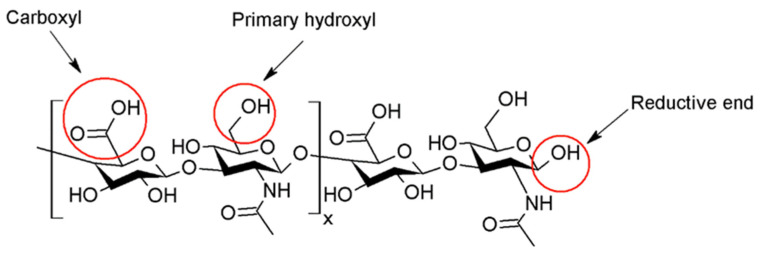
Chemical structure of native HA and functional groups used for its modification.

**Figure 2 polymers-14-04833-f002:**
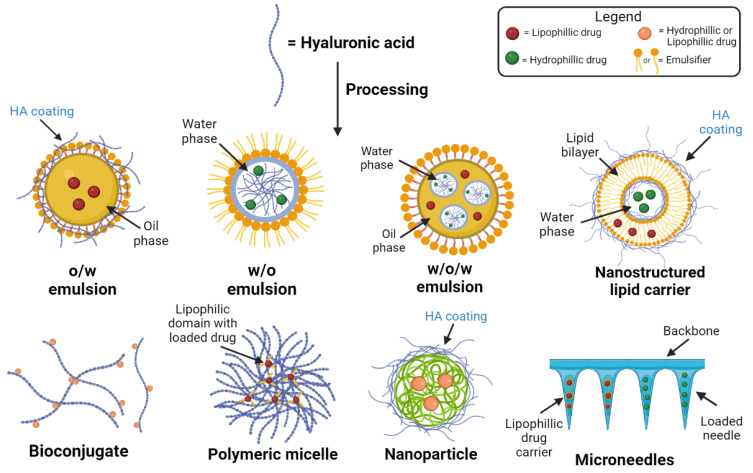
Drug delivery formulations using HA and its derivatives.

**Figure 3 polymers-14-04833-f003:**
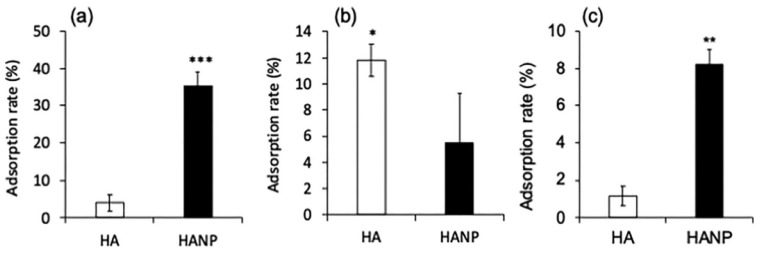
Adsorption of nanoparticulated-HA and HA on (**a**) phospholipid liposomes, (**b**) keratin, or (**c**) SC lipid liposomes. Values represent the mean and standard deviation (SD) of three independent results * *p* <0.05; ** *p* <0.01; *** *p* <0.001. Copyright 2018, with permission from Elsevier.

**Figure 4 polymers-14-04833-f004:**
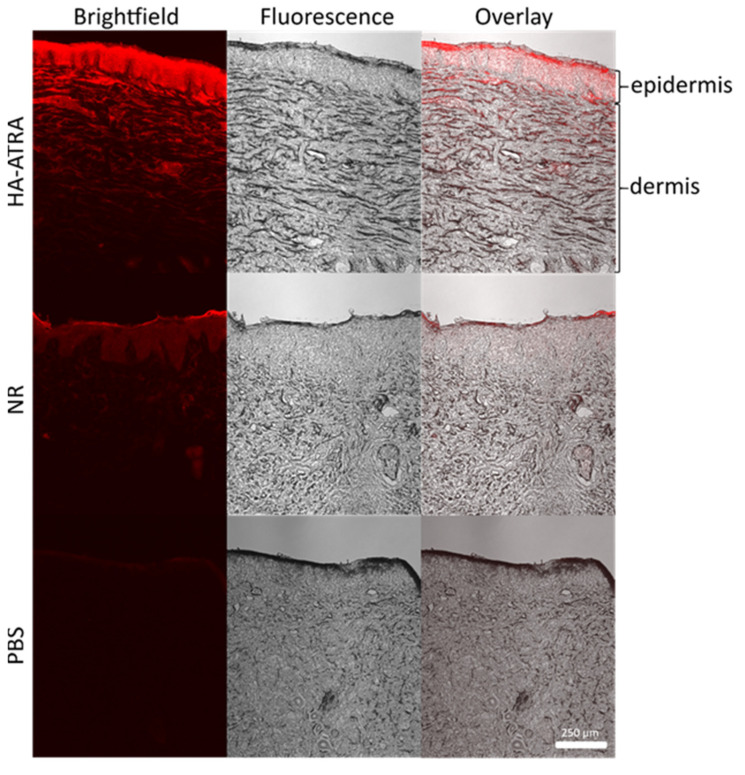
Skin penetration was followed for 24 h in Franz diffusion cells. The amphiphilic derivative made of trans-retinoic acid grafted to hyaluronan was loaded with Nile Red (NR), NR dispersed in PBS and PBS were used as control [6]. Copyright 2020, with permission from Elsevier.

**Figure 5 polymers-14-04833-f005:**
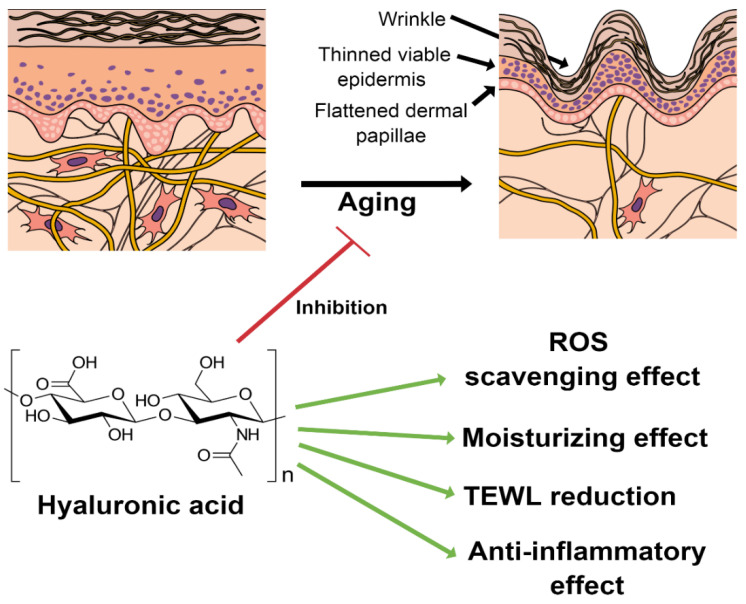
The results obtained after regular use of products containing HA, which demonstrated its role as an active ingredient [81,82,83,84].

**Figure 6 polymers-14-04833-f006:**
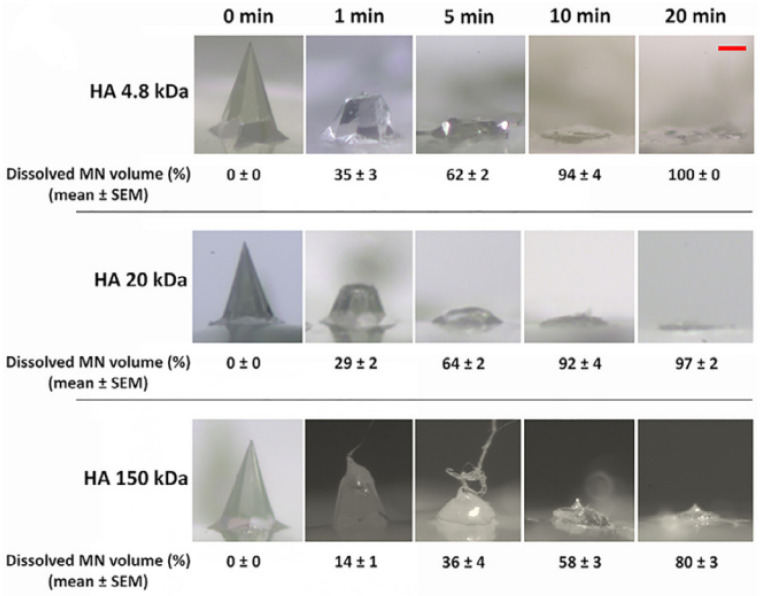
MNs dissolution in ex vivo human skin. Representative bright-field images (5) of 10% (*w*/*v*) HA 4.8 kDa, 20 kDa and 150 kDa microneedle before application on the skin (0 min) and after 1, 5, 10, and 20 min dissolution in ex vivo human skin. Scale bar 100 µm [109]. Copyright 2020, with permission from Elsevier.

**Table 1 polymers-14-04833-t001:** Selected examples describing biological properties after the incorporation of HA in a formulation describing in vitro, in vivo, and clinical data.

Status	Mw (kDa)	Type of Formulation	Advantages Given by the Presence of HA in the Formulation.	Ref.
In vitro/probands	15	Cream	Synergic activity of retinoids and HA.	[6]
In vivo (clinical).	HMW-HA (unknown)	Gel	Combining HA and diclofenac for treatment of atopic dermatitis in high-risk transplant patients, almost half of them remained free of lesions two years post-treatment.	[7]
In vitro and in vivo	<10	Transfersomes	Changed the lipid fluidity	[16]
In vitro	5, 100, 1000	Hydrogel	Occluded the stratum corneum	[22]
In vitro and in vivo (clinical)	A mixture of 50 and 1000.	Serum	Demonstrated no irritation in vitro. Boosted skin hydration.	[27]
In vitro	10	Reverse micelle	Changed the lipids conformation increasing permeation of actives	[47]
Ex vivo	10	Liposomes	Boosted penetration by using amphiphilic HA	[48]
In vitro	<1000	Nanoparticulation	Increased skin permeation more than chitosan	[49]
Ex vivo		Nanoparticulation	Increased both encapsulation efficiency and permeation efficiency of actives	[50]
In vitro and in vivo	300	Nanostructured lipid carriers	Accumulation of the drug on the upper layer of the skin, increased and prolonged anesthetic effect.	[51]
In vivo (clinical)	Crosslinked HA	Serum	Retardance of skin ageing	[52]
In vivo (clinical), ex vivo (human skin explants/biopsies), and in vitro.	unknown	Cream and cream-gel	Significant wrinkle reduction after 28 and 56 days. Pro-collagen and overexpression of HA	[53]
In vitro	150	Ethosomes	Increased penetration efficiency	[54]

**Table 2 polymers-14-04833-t002:** Evaluation parameters of HA-containing micro and nanoemulsions (* nd—zeta potential or size were undefined in the text).

Preparation Method	Active	Mw (kDa)	Application	Size (nm)	Zeta Potential (mV)	Most Relevant Findings	Ref.
High-speed stirring	Azelaic Acid	Unknown	Whitening effect	~419 ± 23	10.9 ± 0.44	Boosted drug penetration	[5]
Ultra-turax^®^ homogenization/second homogenization	*Prunus padus* extract	Unknown	Whitening effect	360–430	nd *	Enhanced anti-wrinkle and whitening effects	[86]
Stirring	Collagen	LMW-HA, Mw < 10 and HMW-HA, Mw ≥ 1000.	Skin regeneration	LMW (72.1 ± 5 to 120.8 ± 4.2) HMW (19.7 ± 3.8 to 85.3 ± 6.1)	nd *	Allowed skin penetration of HA/collagen as a function of HA concentration	[87]
Ultra-turax^®^ homogenisation	*Pterodon pubescens* oil	Unknown	Anti-inflammatory	16–22	−14.2 ± 0.4 to −33.7 ± 1.9	Improved spreading and increased stability.	[88]
Vortexing	*Castor oil*	HMW-HA 1500–1800 LMW-HA 21–40	Cosmetics, sensorial study	nd *	nd *	Molecular weight affects the stringiness of the emulsion.	[89]

**Table 3 polymers-14-04833-t003:** Nanoparticulation of HA for topical delivery (nr * means unreported in the corresponding manuscript).

Formulation	Preparation Method	HA Mw (kDa)	Active	Application	EE (%)	Loading Capacity (%)	Size (nm)	Zeta Potential (mV)	Most Relevant Findings	Ref.
Ethosomes	Stirring/sonification	240	Curcumin	Psoriasis	0.1	nr *	~200	−30	Native HA-coated phospholipid vesicles reduced drug leakage, improved stability, and allowed the slow release of the loaded drug	[14]
Transfersomes	Thin film hydration technique/High pressure homogenization	nr *	Epigallocatechin-3-gallate (EGCGE)	UV radiation-protective	Up to 76.53 ± 2.68 and up to 48.57 ± 4.53 for HA	nr *	101.2 ± 6.0	−44.8 ± 5.24	Native HA-based transferosomes capable of squeezing through the intercellular spaces of the SC. Enhanced EGCGE permeation. Enhanced free radical-scavenging properties and negligible cell toxicity.	[18]
Liposomes	Thin lipid film hydration and extrusion	10	2,3-Di-(4-methoxyphenyl)-quinoxaline	Antileishmanial drug delivery	20–60	2.36 ± 0.34 (HA), 1.53 ± 0.35 (HA-Chol)	214.1 ± 4.3 (HA) and 238.1 ± 6.1 (HA-Chol)	−53.7 ± 1.8, −40.7 ± 5.7	Amphiphilic HA-Chol derivative reached deeper layers of the skin due to higher affinity than native HA.	[48]
Nanostructured lipid carriers	Lipid melt emulsification/solvent injection	300	Bupivacaine	Antinociceptive	88.9 ± 3.1	1.7 ± 0.2	154.6 ± 5.1	−40	Improved percutaneous penetration for the amphiphilic derivative HA-PEG-LOA (BPV/NLCs than non-modified BPV/NLCs or free BPV.	[51]
	Solvent diffusion method	3	Ropivacaine/Dexmedetomidine	Anesthesia/antinociception effect	90/88	16 ± 3	100	−30	Increased skin penetration by using the amphiphilic HA-PEG-DSPE derivative.	[63]
Nanoparticles	High-pressure homogenization–evaporation method	100	Betamethasone	Atopic dermatitis	73.43 ± 6.7 to 87.43 ± 9.1	27 ± 4.12 to 35 ± 6.39	279 ± 12 to 554 ± 23	61.5 ± 4.8 to 44.5 ± 4.6	pH-controlled and sustained release.	[50]
	High-pressure homogenization– evaporation method	100	Tacrolimus	Anti-dermatitis	84.11 ± 4.93	29.34 ± 2.13	216 ± 16 to 389 ± 26 with HA%	51 ± 4.67 to 34 ± 5.23	Native HA improved thermal stability and decreased crystallinity. Sustained release. Possibility of targeting.	[65]
Niosomes	Thin-film hydration method.	30	*Centella asiatica* extract	Herbal therapy for skin disorders (via eczema, wound healing)	41.19 ± 0.61 to 73.54 ± 0.32	4.36 to 9.61	100–180	−40 to −10	Native HA enhanced bioactivity with potential as anti-psoriasis, eczema, anti-inflammation, or anti-ageing treatments.	[93]
	Thin-film hydration method.	200–400	Curcumin/quercetin	Antioxidant and anti-inflammatory	98.85 ± 0.55 93.13 ± 1.22	2 and 2.68	260.4 ± 6.6	−34.97 ± 1.50	Improved nanovesicle stability and allowed selective targeting. Native HA does not change the EE.	[94]
Nanoparticulated gel	Lipid fusion, and ultrasound	300	Caffeic acid	Antioxidant	87.8 ± 5.2	1.7 ± 0.05	230 ± 14	nr	Increased spreadability of the formulation	[95]
Polymeric micelles	Dialysis	44 and <10	Curcumin	Antioxidant and anti-inflammatory	43.10 ± 5.4	5.62 ± 1.7	172.6 ± 11.4	−33.71 ± 0.45 mV	Bioconjugation with HA allowed the solubilization of curcumin in medium.	[20]
	Solvent-evaporation method	15	Coenzyme Q10	Cosmetic ingredient	85.4 ± 0.4	9.8 ± 0.8	163	−25.6	The drug-loading capacity is controlled by chemical modification and structure of carrier.	[19,96]
Nanogels	Ionotropic gelation	10	Methotrexate/5-aminolevulinic acid	Psoriasis	75.42/71.94	0.05 and 10	141.43 ± 0.47	31.59 ± 0.44	Complexed chitosan/HA complex reduced systemic toxicity of the drugs.	[97]

**Table 4 polymers-14-04833-t004:** Recent advances in dissolvable MNs of HA and its derivatives include processing, needle length, and significant outcomes (* nd means non-described in the corresponding manuscript, h stands for height, and b means base.

Preparation Method	Active	Mw (kDa)	Application	Length (μm)	Most Relevant Findings	Ref.
Two-step micromolding	Methotrexate (MTX)	10	Psoriasis	650 ± 19 (h), 220 ± 7(b)	The mechanical properties were enough to pierce the skin and reached dermis	[4]
Micro-molding	5-Aminolevulinic acid	10	Photodynamic therapy	650 h	LMW-HA effectively protects the active from degradation	[8]
Micro-molding	Ovalbumin (OVA)	7 and 200–500	Immunization	550 (h), 300 (b)	Stronger immune response by HA	[64]
Centrifugal lithography	Adenosine	HMW-HA 800 LMW-HA 39	Cosmetics	402.5 ± 12.9 81.2 ± 9.3	HMW-HA presented better skin improvement effects, lower depth of wrinkles, and increased elasticity and dermal density	[107]
Casting-molding and centrifugation	Rutin	403	Cosmetics	550 (h) 200 (b)	Combination of bacterial cellulose and HA	[108]
Micro-molding	Ovalbumin (OVA)	4, 20, 150 and 1800	Immunization	300(h)	LMW-HA (150 kDa) showed a higher penetration efficiency (<96%) and dissolved faster than LMW-HA (4.8 kDa) or HMW-HA (1.8 MDa), which was unsuitable for the fabrication	[109]
Two-step casting	Bleomycin	10	Hypertrophic scars	650 h	Fast dissolution ~10 min. Synergic effect of HA/bleomycin inhibited the proliferation of human hypertrophic scar fibroblasts.	[110]
Micro-molding	*Lonicerae flos* extract	nd *	Cosmetics	nd *	Native HA and the plant extract synergism’ improved skin moisturizing properties.	[111]
Micro-molding	Propranolol hydrochloride	10	Infantile hemangioma	1200 (h), 300 (b)	LMW-HA was processed in an obelisk shape, giving higher cutaneous delivery than pyramidal or circular shapes.	[112]
Micro-molding and nanoparticulation		nd *	nd *	600 (h), 250 (b)	Crosslinked HA presented higher mechanical strength due to the enhanced viscosity.	[113]

## Data Availability

Data presented in this study are available on request from the corresponding author.

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
