# Peer review of "Recent Advances of Hyaluronan for Skin Delivery: From Structure to Fabrication Strategies and Applications"

_polymers, 2022, doi:10.3390/polym14224833_

Round 1

Reviewer 1 Report

The authors completed a well-organized review paper with profound details regards to the application of HA in skincare research. This manuscript was impressive and highly qualified for this elite journal.

The authors described skin deliverable HA in three aspects:

1. From the hyaluronan material itself, describe the natural chemistry properties of HA and how it could be utilized.

2. From the application perspective, how HA could benefit skin based on cellular to clinical levels.

3. Based on the fabrication point of view, the authors provided a comprehensive summary of how different formats of HA were produced and how they were applied.

If the authors could prepare a table summarizing the state-of-the-art status of in vitro, in vivo, and clinical studies as you did in Table 2, it would be an excellent benefit for readers who were not familiar with biological experiments. As I considered Figure 3 & 4 were simplified visualization results from only two papers. Let along Figure 5, which is a schematic illustration without evidence.

However, there are a few minor issues the authors may consider correcting.

Line 56-59: Convoluted sentences. As for a scientific research review paper, mentioning business is ok but must have sufficient supplementary discussion or references. I would suggest removing this introductory level business discussion as it is unclear if the subdivided market is solely about HA or microneedles.

Line 100: Perhaps it is unnecessary to add a sub-title as there are no other parallel sections.

some typos: Page 19 Line 1, Page 20 Line 56& 84, Page 23, Line 165: 

Author Response

The authors completed a well-organized review paper with profound details regards to the application of HA in skincare research. This manuscript was impressive and highly qualified for this elite journal.

The authors described skin deliverable HA in three aspects:

  1. From the hyaluronan material itself, describe the natural chemistry properties of HA and how it could be utilized.

  1. From the application perspective, how HA could benefit skin based on cellular to clinical levels.

  1. Based on the fabrication point of view, the authors provided a comprehensive summary of how different formats of HA were produced and how they were applied.

If the authors could prepare a table summarizing the state-of-the-art status of in vitro, in vivo, and clinical studies as you did in Table 2, it would be an excellent benefit for readers who were not familiar with biological experiments. As I considered Figure 3 & 4 were simplified visualization results from only two papers. Let along Figure 5, which is a schematic illustration without evidence.

Answer: a table was prepared.

The figure was corrected, and 4 important citations were added in the text.

However, there are a few minor issues the authors may consider correcting.

Line 56-59: Convoluted sentences. As for a scientific research review paper, mentioning business is ok but must have sufficient supplementary discussion or references. I would suggest removing this introductory level business discussion as it is unclear if the subdivided market is solely about HA or microneedles.

Answer: The convoluted sentences were deleted.

Line 100: Perhaps it is unnecessary to add a sub-title as there are no other parallel sections.

Answer: The sections were corrected

some typos: Page 19 Line 1, Page 20 Line 56& 84, Page 23, Line 165:

Answer: All the article was revised and corrected.

Reviewer 2 Report

1-    The reference distribution in the first paragraph of the introduction is not acceptable. Try to use a relevant reference for every fact that is discussed.

2-    Make a schematic figure to show the function of HA in the skin as an anti-aging factor.

3-    What is the best HA loading percentage for a better result on the skin?

4-    The following references can be used in the manuscript to improve the discussion part:

Sabbagh, F., & Kim, B. S. (2022). Microneedles for transdermal drug delivery using clay-based composites. Expert Opinion on Drug Delivery19(9), 1099-1113.

Sabbagh, F., & Kim, B. S. (2022). Recent advances in polymeric transdermal drug delivery systems. Journal of Controlled Release341, 132-146.

Author Response

1. The reference distribution in the first paragraph of the introduction is not acceptable. Try to use a relevant reference for every fact that is discussed.

Answer: The references were included according to the same citations.

2. Make a schematic figure to show the function of HA in the skin as an anti-aging factor.

The function is complex and hard to represent in a single scheme. We have included Table  2 to enunciate the clear advantages.

In cosmetic formulations, hyaluronic acid has the function of a viscosity modifier and/or a skin conditioning agent. HA is mainly used in anti-ageing cosmetic products. LMW-HA has the ability to enhance the level of moisture of the skin and expedite regeneration. HMW-HA forms a viscoelastic film when applied onto the skin and has a moisturizing effect. The main action of the HMW-HA polymer is film forming and it reduces evaporation of water from the skin and thus possesses an occlusive effect. Additionally, HMW-HA, Medium molecular weight (MMW-HA), and LMW-HA hygroscopic properties justify the ability to maintain skin hydration

3. What is the best HA loading percentage for a better result on the skin?

Answer, the best answer to this question is nobody knows.  HA is not really loaded but it is incorporated in a formulation either as a crosslinked gel or as a coating. Thus, loading and results are not related concepts.

The percentage varies for micelles 2% (Kupper et al, Line 592 and 552). While nanoparticles 0,2% (Zhuo et al. and 566)

4-    The following references can be used in the manuscript to improve the discussion part:

Sabbagh, F., & Kim, B. S. (2022). Microneedles for transdermal drug delivery using clay-based composites. Expert Opinion on Drug Delivery, 19(9), 1099-1113.

Answer: Thank you so much for your recommendation. However, this review does not include clays, neither composites. Thus, the citation is not relevant. According to the clinics and preclinic studies analysed in this work. Only native HA or crosslinked by short linkers are currently marketed products, but clays cannot be applied in vivo (intradermal) due to expected toxicity.

Sabbagh, F., & Kim, B. S. (2022). Recent advances in polymeric transdermal drug delivery systems. Journal of Controlled Release, 341, 132-146.

Answer: Thanks for your recommendation, but, this citation does not include the role of hyaluronan. Actually, the only reference where hyaluronan is included is as a part of a thin film-based composite, which is obtained by solvent casting. This method is not applicable. Thus, it is considered out of the scope.